# LWR-Net: Robust and Lightweight Place Recognition Network for Noisy and Low-Density Point Clouds

**DOI:** 10.3390/s23218664

**Published:** 2023-10-24

**Authors:** Zhenghua Zhang, Guoliang Chen, Mingcong Shu, Xuan Wang

**Affiliations:** 1School of Environment Science and Spatial Informatics, China University of Mining and Technology, Xuzhou 221116, China; 13685196078@163.com (Z.Z.);; 2State Key Laboratory of Information Engineering in Surveying, Mapping and Remote Sensing, Wuhan University, Wuhan 430079, China

**Keywords:** place recognition, point cloud retrieval, lightweight 3D learning, noisy and low-density point cloud

## Abstract

Point cloud-based retrieval for place recognition is essential in robotic applications like autonomous driving or simultaneous localization and mapping. However, this remains challenging in complex real-world scenes. Existing methods are sensitive to noisy, low-density point clouds and require extensive storage and computation, posing limitations for hardware-limited scenarios. To overcome these challenges, we propose LWR-Net, a lightweight place recognition network for efficient and robust point cloud retrieval in noisy, low-density conditions. Our approach incorporates a fast dilated sampling and grouping module with a residual MLP structure to learn geometric features from local neighborhoods. We also introduce a lightweight attentional weighting module to enhance global feature representation. By utilizing the Generalized Mean pooling structure, we aggregated the global descriptor for point cloud retrieval. We validated LWR-Net’s efficiency and robustness on the Oxford robotcar dataset and three in-house datasets. The results demonstrate that our method efficiently and accurately retrieves matching scenes while being more robust to variations in point density and noise intensity. LWR-Net achieves state-of-the-art accuracy and robustness with a lightweight model size of 0.4M parameters. These efficiency, robustness, and lightweight advantages make our network highly suitable for robotic applications relying on point cloud-based place recognition.

## 1. Introduction

Accurate localization plays a crucial role in the operation of autonomous vehicles and robots. It not only enables confident navigation and obstacle avoidance but also facilitates efficient task execution. Localization serves as the foundation for enhanced mobility and automation. One specific aspect of localization is place recognition, which involves searching a database of geo-tagged scene data to find the descriptor most similar to the query scene. This becomes particularly important when localizing in environments where reliable GPS signals are unavailable [1] or when using SLAM systems that require loop closure [2,3,4]. While vision-based methods can be sensitive to factors like illumination, camera field of view, and viewing orientation [5], lidar-based solutions offer greater robustness to different lighting conditions and seasonal changes. However, real-world environments present challenges that can degrade lidar-based place recognition performance. Factors such as varying point density, noise, and computational resource limitations for robotic applications [6] must be addressed when designing robust and efficient lidar-based place recognition methods.

The advancement of hardware has dramatically contributed to the development of deep learning in 3D vision [7,8], leading to a shift from traditional hand-crafted features [9,10,11] to learning-based methods in place recognition. These learning-based architectures can be broadly categorized into point-based and voxel-based methods based on their data representations. Point-based methods treat the point cloud as a sequence of independent points or utilize information from the local neighborhood of each point. For instance, PointNetVLAD [12] combines PointNet [13] to extract individual point features and NetVLAD [14] to generate a global descriptor. Other methods, such as PCAN [15], LPD-Net [16], and DAGC [17], improve performance by employing graph-based or attention-based structures to aggregate more informative neighborhood features. Recently, EPC-Net [18] was proposed to enhance efficiency by introducing lightweight ProxyConv and G-Vlad modules. On the other hand, voxel-based methods discretize the point cloud into a regular grid map and utilize 3D convolution to learn features hierarchically. The pioneering voxel-based method, MinkLoc3D [19], employs sparse 3D convolution [20] to extract local features in the voxel domain and uses Generalized Mean (GeM) pooling to aggregate the global descriptor. Minkloc++ [21] fuses the monocular image as an extra source data to promote accuracy further. TransLoc3D [22] further incorporates an adaptive receptive field module to address size variations in complex scenes.

While these methods have demonstrated promising results, they have limitations when applied to complex real-world scenes. Most existing methods are trained and evaluated using point clouds with low noise and sufficient point density, leaving the performance relatively unexplored in highly noisy and low-density conditions. Moreover, many of these methods heavily rely on sophisticated local geometric extractors, leading to memory access overhead and large parameter sizes (Figure 1), thereby decreasing efficiency and capacity in real-world applications. Consequently, a continued demand exists for an efficient and lightweight lidar-based place recognition method capable of effectively handling noisy and low-density point clouds.

To address the challenges above, we propose a lightweight retrieval network called LWR-Net, explicitly designed for efficient point cloud-based place recognition in noisy and low-density conditions. LWR-Net follows a simple two-stage architecture but delivers highly competitive performance. In the feature extraction stage, we introduce a fast dilated sampling and grouping module to collect features from local regions efficiently. These local features are then hierarchically aggregated using a feed-forward residual multi-layer perceptron (MLP) structure. To enhance robustness and generalization, we incorporate a lightweight attentional weighting module during the encoding process, which improves the features from a global perspective. In the feature fusion stage, we utilize Generalized Mean (GeM) pooling [23] to aggregate the global descriptor into a fixed 256-dimensional representation. Due to its simplicity and effectiveness, LWR-Net has a significantly smaller model size compared to existing methods (Figure 1). We conducted extensive experiments using the Oxford robotcar dataset [24] and three in-house datasets [12] to validate the performance of our network. Despite its lightweight design, the results demonstrate that LWR-Net achieves state-of-the-art accuracy and runtime efficiency while exhibiting higher robustness to noisy and low-density conditions. Furthermore, our network demonstrates better generalization capabilities when dealing with environmental changes.

## 2. Methods

We proposed a simple yet effective architecture called LWR-Net for accurately retrieving scene point clouds with a low parameter size. LWR-Net follows a two-stage pipeline architecture, as shown in Figure 2, which includes local feature extraction (Section 2.1) and global descriptor generation (Section 2.2). The implementation details of LWR-Net are described in Section 2.3. Let $P\in R^{N\times 3}$ be a raw point cloud with N points, each with a 3-dimensional feature representing spatial coordinates. LWR-Net takes P as input and outputs the global descriptor $G\in R^{256}$ for efficient retrieval.

### 2.1. Local Feature Extraction

In the local feature extraction stage, the network needs to explore local geometric information directly from the large-scale point cloud. The feature extractor used in this stage has two major requirements: (1) it can accurately sense the local information in the scene to assist in aggregating more distinctive global features; (2) it must be lightweight to keep the memory and computation efficiency, which means no sophisticated or heavy operations should be used in the feature extractor. To meet the demand, we designed the feature encoding layer consisting of three parts: fast dilated sampling and grouping, residual-based encoding, and attentional weighting.

#### 2.1.1. Fast Dilated Sampling and Grouping Module

Sampling and grouping are essential to aggregate local neighborhood features for point-based learning structures. Usually, the first step is to sample a subset of points $P_{s}\in R^{N_{s}\times 3}$ representing the local regions’ centroid, then constructing a local region set by finding neighborhood points around the centroid. We chose random sampling in our method, which had the computational complexity of O(1), and the sampling process is agnostic to the point number, which fits our demand for efficiency.

For the grouping process, we chose the K-nearest neighborhood (KNN) algorithm to group k neighbor points of the sampled centroid. To accelerate the speed, we used the KD-tree structure [25] to reduce the computational complexity of the ordinary KNN algorithm ($O\left(N\right)$ to O(logN)) [26], which further improves the efficiency of the network.

Since the random sampling algorithm is sensitive to uneven point density distributions, areas with high point density tend to have more sampled points, adversely affecting the perception of the scene. To alleviate this issue, we incorporated a strategy called dilated sampling. This strategy first expands the number of sampling point $N_{s}$ by a dilated ratio σ, then calculates the average distance between neighboring points and sampling points. This distance can indicate point density within each local region since we utilized the KNN algorithm for grouping. We sorted the sampling regions in descending order based on the calculated distance. We then selected the top $N_{s}$ regions as the final output. This strategy helps suppress excessive sampling points from being concentrated in high-density areas. The input to this module is a point set of size N×3; each point corresponds to a feature vector, denoted as $F_{in}\in R^{N\times C_{in}}$ ($C_{in}=3\;for\;the\;first\;layer)$, and the output is groups of point sets and features of size $N_{s}\times k\times 3$ and $N_{s}\times k\times C_{in}$. Each group represents a local region of the input point cloud. The grouped feature of the local region is denoted as $F_{region}^{group}$.

#### 2.1.2. Residual MLP Encoding Module

We designed a simple feed-forward residual MLP module to learn the local neighborhood representation. This module takes $F_{region}^{group}$ as the input, and the output is the global representation of the sampled region, denoted as $F_{region}^{global}$ of size $N_{s}\times c_{out}$. The key operation of the encoding process can be formulated as follows:(1)$$F_{region}^{global}=\Phi_{global}^{1D}(Atten(Max(\Phi_{local}^{2D}({{MLP}_{\left[c_{in},c_{out}\right]}(F}_{region}^{group})))))$$
where ${MLP}_{\left[c_{in},c_{out}\right]}$ is the basic encoding block to increase the feature dimension from $c_{in}$ to $c_{out}$. Max(·) is the max-pooling operation to aggregate features into a global representation. Atten(·) represents the attentional weighting module, which is introduced in the following subsection. Φ(·) denotes the residual MLP block; the basic structure is shown in Figure 2. The residual MLP block has two versions: local residual block ($\Phi_{local}^{2D}(\cdot )$) and global residual block ($\Phi_{global}^{1D}(\cdot )$). The local residual block is designed to learn shared weights from local regions’ encoded features; the global residual block is utilized for encoding deep aggregated features. Since this module only leverages MLPs and residual connections, no sophisticated structures are introduced; it is highly efficient and naturally invariant to permutation.

Features encoded by residual blocks only contain regional geometric information; we embedded an attentional weighting module before $\Phi_{global}^{1D}$ to learn the long-term relevance of different local regions and strengthen the feature from a global perspective. As illustrated in Figure 2, we designed a grouped self-attention as the core operation in this module, which is a lightweight but effective version of the attention mechanism [27]. The input to this module is the feature map of local regions after aggregation, denoted as $F_{aggre}\in R^{N_{s}\times C}$, we first generated the query, key, and value feature map using the following formulas:(2)$$Q_{g}=W_{q}^{g}(F_{aggre}),K_{g}=W_{k}^{g}(F_{aggre}),V=W_{v}(F_{aggre})$$
where $W_{q}^{g}\left(\cdot \right)$ is a group-wise 1×1 convolutional layer, which divided the query feature map into G groups, denoted as $\left\{Q_{g}\in R^{N_{s}\times \left(\frac{C}{G}\right)}|g=1,\ldots ,G\right\}$, the key feature map $K_{g}$ is divided into G groups by $W_{k}^{g}\left(\cdot \right)$ likewise.

Meanwhile, we adopted another 1×1 convolutional layer, denoted as $W_{v}\left(\cdot \right)$, to generate the value feature map $V\in R^{N_{s}\times C}$. For each group of query map and key map, the attentional weighting matrix is calculated, and the final attention map is obtained by summing up the weighting matrix of G groups, which is formulated below:(3)$$W=\sum_{g=1}^{G}{Q_{g}K}_{g}^{T}$$
where $W\in R^{N_{s}\times N_{s}}$ is the attentional weighting matrix. The resulting feature map of the grouped self-attention layer, denoted as $F_{atten}\in R^{N_{s}\times C}$, is written as:
(4)$$F_{atten}=Softmax\left(\frac{W}{\sqrt{C}}\right)V+F_{aggre}$$

By dividing the channel axis into G groups, the attentional weighting module can obtain a more detailed weighting matrix and decrease the computational complexity of the ordinary self-attention mechanism from $O(m^{2}\cdot C)$ to $O(m^{2}\cdot \left(\frac{C}{G}+G\right))$, simplifying the network and further enhancing its efficiency. Afterward, $F_{atten}$ is further encoded by $\Phi_{global}^{1D}$ to generate $F_{region}^{global}$.

### 2.2. Global Descriptor Generation and Loss Function

Since the commonly used NetVLAD is inefficient and memory-intensive, we aggregated the global descriptor using the GeM pooling function [23]. GeM pooling is a generalization of global max pooling and global average pooling, which is highly efficient and can be formulated as:(5)$$G=\left[f_{1},f_{1},\ldots ,f_{C}\right],\;f_{C}=\frac{1}{\left|F_{region}^{global},C\right|}\sum_{x\in F_{region}^{global},C}{\left(x^{p_{k}}\right)}^{\frac{1}{p_{k}}}$$
where $G\in R^{C}$ is the final descriptor, and $p_{k}$ is a learnable control parameter.

The loss function for our network aims at minimizing the feature distance between structurally similar point clouds while maximizing the feature distance between structurally dissimilar point clouds. We adopted triplet margin loss [12] to train our network:(6)$$L\left(G_{i},G_{i}^{p},G_{i}^{n}\right)=max\{d\left(G_{i},G_{i}^{p}\right)-d\left(G_{i},G_{i}^{n}\right)+m,0\}$$
where $G_{i}$ is the global descriptor of the query point cloud; $G_{i}^{p}$ and $G_{i}^{n}$ are the positive and negative samples, respectively; $d\left(x,y\right)$ define the Euclidean distance function between x and y; m is a constant parameter giving the margin. To build the training tuples, we used batch hard negative mining proposed in [19], which decreases the training time from days to hours while maintaining effectiveness.

### 2.3. Implementation Details

We applied a very shallow network by hierarchically using two feature encoding layers to gather local features of the input point cloud; the number of sampling points of each layer was 1024 and 256, and the number of neighborhood points was 32 and 16. The dilated ratio σ was set to 1.2 in the experiment. When feeding the point cloud into LWR-Net, the neuron size of the obtained feature maps in two layers were 1024×64 and 256×256. For the attentional weighting module, the number of groups was set to 2 and 4 in each grouped self-attention. We adopted the Adam optimizer with the learning rate of 1 × 10^−3^ for 60 epochs, which decayed by 0.1 at epochs 30 and 50. All experiments were conducted on an Intel i7-9700k CPU using a single NVIDIA RTX2070s graphic card.

## 3. Experiments

### 3.1. Experimental Setting

#### 3.1.1. Benchmark Datasets and Preprocessing

Following previous works in place recognition, we used the modified Oxford robotcar dataset [24] and three in-house datasets [12] to evaluate our model. The Oxford robotcar dataset was created using a Sick LMS-151 lidar scanner mounted on a vehicle, traversing a 10 km route through central Oxford repeatedly to capture data. The three in-house datasets, including a university sector (U.S.), a residential area (R.A.), and a business district (B.D.), were obtained by the National University of Singapore. These datasets were captured using a Velodyne-64 lidar scanner, covering continuous trajectories of 10 km, 8 km, and 5 km, respectively. Each submap represents a unique local area of the region and is tagged with a UTM coordinate with respect to GPS/INS reading. To generate training tuples, point cloud pairs with a distance of less than 10 m were defined as positive pairs, while more than 50 m were defined as negative pairs. Each submap was preprocessed using the same standard pipeline to learn geometric features better; the non-informative ground was first removed in each submap, and then the resulting point cloud was uniformly down-sampled to 4096 points. The point cloud of each submap was finally shifted and rescaled to be zero mean and inside the range of [−1, 1].

#### 3.1.2. Comparison Methods and Evaluation

We compared the proposed network with a series of advanced methods in the experiments, including PointNetVLAD [12], PCAN [15], LPD-Net [16], SOE-Net [28], EPC-Net [18], Minkloc3D [19], NDT-Transformer [29], MinkLoc++ [21], PPTNet [30], MinkLoc-v2 [31], and SVT-Net [32]. All these methods are trained on the Oxford robotcar dataset to produce the global descriptor of size 256. All these methods were evaluated using their author’s released code and pre-trained model except the NDT-Transformer. Since the NDT feature generation part is missing in the NDT-Transformer’s source code, we took the results reported by other works following the same evaluation protocol. It is worth noting that ASVT-Net is a simplified version of SVT-Net, with a smaller model size while maintaining accurate performance.

For the performance evaluation, we assumed that the query point cloud is successfully recognized if at least one of the top-K retrieved database clouds is within 25 m from the ground truth position of the query. Therefore, we followed the same evaluation metrics used in [12], which is the average Recall@K indices (including Recall@1%, Recall@1, and Recall@25), defined as the percentage of correctly recognized queries. We also measured the average running time and parameter number to evaluate the efficiency and model size.

### 3.2. Place Recognition Results

The evaluation results on the standard Oxford robotcar dataset are illustrated in Table 1. We can see that the model size of the existing methods is quite different. Our method can achieve high accuracy while maintaining computational efficiency. The model size is reduced by more than 50%, which indicates that LWR-Net achieved comparable accuracy with less than half the parameters of the existing methods. Despite MinkLoc-v2 showcasing better accuracy in this experiment, LWR-Net demonstrates approximately twice the speed and a six times model size reduction. Furthermore, as we observed in the subsequent experiments, LWR-Net exhibits stronger robustness under point cloud sparsity and noise intensity variations and superior generalization ability.

### 3.3. Robustness Analysis

#### 3.3.1. Robustness to Point Cloud Density

Firstly, we investigated the influence of point density. We randomly down-sampled the test set with different point numbers to simulate the density changes. We chose Minkloc++, MinkLoc3D, PPT-Net, SOE-Net, EPC-Net, MinkLoc-v2, and SVT-Net to compare in this experiment, and we assume the method fails if the recall indices are less than 30%. Based on the trained model, we evaluated the retrieval accuracy under different point density settings; Figure 3 shows the results. As we can see, EPC-Net, SOE-Net, and SVT-Net are very susceptible to point density changes; MinkLoc3D, MinkLoc++, and MinkLoc-v2 are voxel-based architecture; and the sparsity of point cloud inevitably influences them by changing the distribution of voxel grids. LWR-Net is designed based on random sampling, making our network naturally adaptable to point sparsity. As a result, LWR-Net is more robust to the point density changes than other methods in this experiment.

Figure 4 shows the average recall curves of LWR-Net, PPT-Net, MinkLoc++, and MinkLoc-v2 for the top 25 matches under influences of different point densities, which also proves that our method is more robust than other comparison methods to the low-density environment.

#### 3.3.2. Robustness to Point Cloud Noises

We developed a simulation strategy to add random outlier noise to the scene point cloud, which includes the following steps: (1) Generate a random noise location within the scene range. (2) Find the nearest neighbor of the generated noise location and calculate the distance. If the distance is less than the set threshold (5 cm in the experiment), we assume this location is too close to the original surfaces in the point cloud (making it more like a measurement noise), and we re-generate a new location according to the previous step; otherwise, we add it into the scene point cloud as an outlier noise point. (3) Loop executes steps 1 and 2 until the number of outlier noise points exceeds the set threshold. We compared LWR-Net to PPT-Net, MinkLoc3d, MinkLoc++, SOE-Net, EPC-Net, MinkLoc-v2, and SVT-Net in this experiment, and we still set 30% as the threshold to determine whether the method failed. Figure 5 shows the performance of each method under the influences of random outlier noise intensity. We can see that SOE-Net, SVT-Net, and EPC-Net are comparably more vulnerable to random outlier noises. Although PPT-Net, MinkLoc3D, MinkLoc-v2, and MinkLoc++ have a specific resistance to continuously increased noise, LWR-Net is the most robust method in this experiment. Even if the number of random outlier points reaches a quarter of the total point number, our method can still provide reliable accuracy.

Figure 6 shows the changes in recall curves of LWR-Net, PPT-Net, MinkLoc++, and MinkLoc-v2 for the top 25 retrieval results under the influence of random outlier noises, where our LWR-Net is superior to other methods. This further demonstrates that our method is more suitable for application in real-world environments with random outlier noise influences.

Furthermore, we tested the robustness of LWR-Net to the measurement noise. We randomly jittered the points in the test point clouds by noise sampled from $N(0,\sigma^{2})$ and clipped to $\left[-0.05\;m,0.05\;m\right]$ on each axis; we adjust the variance from 0.01 to 0.05 to continuously increase the noise intensity. We used the trained model on clean data and tested it on noisy data. Figure 7 shows that even if the measurement noise increased, the impact on our method’s accuracy is within 1%, proving our method’s robustness to measurement noise.

### 3.4. Generalization Analysis

In order to verify the generalization ability of each method, we performed a cross-dataset experiment using three in-house datasets. We used the trained model on the Oxford robotcar dataset and directly evaluated the performance on three in-house datasets. Table 2 shows the results, demonstrating that our method is more generalized and has a stronger discrimination ability under the influence of environmental changes.

We further tested the robustness of our method on three in-house datasets; note that we never trained our approach on the in-house datasets. Figure 8 shows the results. Even though LWR-Net is slightly affected by point density and noise intensity changes, our method can still maintain certain robustness in the new environment.

### 3.5. Ablation Study

We conducted experiments to verify the effectiveness of our model design. We eliminate different network components and generate multiple ablated versions of the LWR-Net, including the following: (1) LWR-Core: the attentional weighting module was eliminated and only preserved the core structure of the network. (2) LWR-WithoutRes: we used the ordinary MLP block with the same channel dimension to replace the residual MLP encoding module in the network. (3) LWR-WithoutResLocal/ResGlobal: since we used two different residual blocks ($\Phi_{local}^{2D}$ and $\Phi_{global}^{1D}$) to learn the local and aggregated features of the region, we separately eliminated $\Phi_{local}^{2D}$ and $\Phi_{global}^{1D}$ to test the impact of the residual block at different positions; (4) LWR-WithoutDS: the ordinary random sampling algorithm was used to replace the dilated sampling algorithm in the architecture. We trained each network on the Oxford robotcar dataset, and Table 3 shows the results.

Based on the experimental results, we can observe the following facts: (1) the residual connection of the encoding module can effectively improve the performance of the network; (2) removing either $\Phi_{local}^{2D}$ or $\Phi_{global}^{1D}$ will cause a drop in accuracy, demonstrating that our strategy for using two residual blocks is correct; (3) the dilated sampling structure and attentional weighting module can further promote the retrieval accuracy of the network, but the parameter number of the attentional weighting module is 0.1 million. As a result, we can further serve the attentional weighting module as an alternative option in real-world applications to adjust the balance between model size and retrieval accuracy.

### 3.6. Discussion

The experiments show that compared to the state-of-the-art place recognition methods, LWR-Net can provide accurate retrieval results and is more robust under various point densities and noise intensity. Moreover, our network has a stronger generalization ability, making it maintain reliable accuracy and robustness under environmental changes. Most importantly, LWR-Net achieves the above performance with only a small number of parameters. The model size of LWR-Net is reduced by more than 50% compared with existing methods, making it more efficient in processing large-scale scene data. Figure 9 shows the retrieval results of our network on the Oxford robotcar dataset and three in-house datasets. As we can see, LWR-Net can accurately recognize the correct place throughout the entire reference database, and the top 3 candidates of retrieval results are structurally similar scenes to the query point cloud.

Figure 10 shows the place recognition results of our network in low-density and noisy environments. Even though the query point cloud is sparse or corrupted with a large amount of noise, LWR-Net can still provide reliable retrieval results and further realize fast localization.

## 4. Conclusions

In this work, we proposed a lightweight network named LWR-Net for point cloud-based place recognition. We designed LWR-Net based on random sampling and residual MLPs and further introduced an attentional weighting module to promote performance. Experimental results have shown that our network outperforms related works in simplicity and efficiency and is also more robust under low-density and noisy situations. Moreover, LWR-Net performs more generalized under environmental changes. Most importantly, LWR-Net achieves the above performance with a small model size (0.4 million parameters), proving that the sophisticated feature extractor may not be crucial for the place recognition network.

Although our network has improved the efficiency and robustness of point cloud retrieval tasks, it still has some limitations. The accuracy and robustness of LWR-Net under low-density environments can be further promoted. LWR-Net proves the effectiveness of random sampling and residual MLPs in designing the place recognition network; future work can further design a more powerful structure under this frame. Since our network is lightweight and robust, another possibility for future work would be to incorporate LWR-Net into SLAM, which can demonstrate its value for robotic applications.

## Figures and Tables

**Figure 1 sensors-23-08664-f001:**
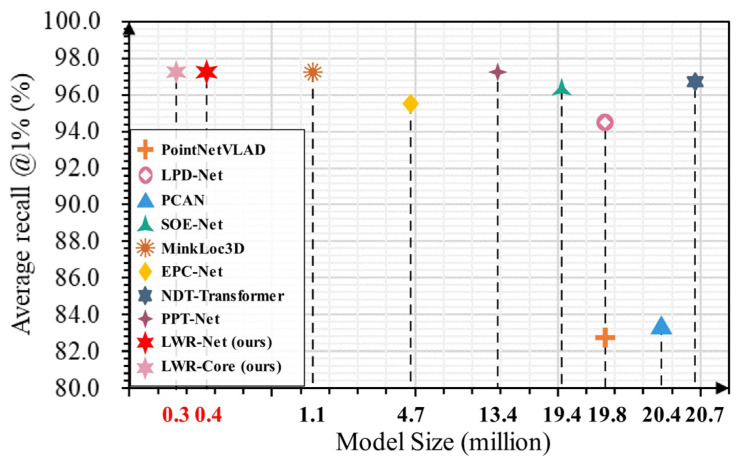
Model size and accuracy comparison of existing lidar-based place recognition methods.

**Figure 2 sensors-23-08664-f002:**
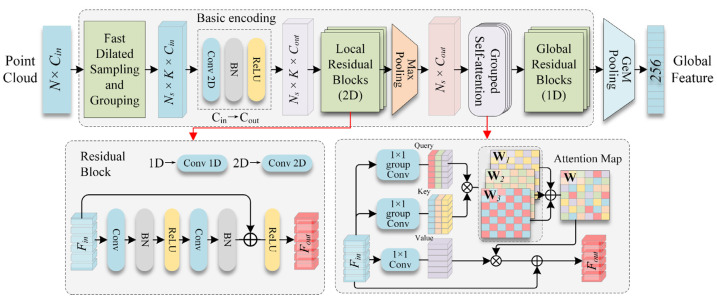
The architecture of LWR-Net.

**Figure 3 sensors-23-08664-f003:**
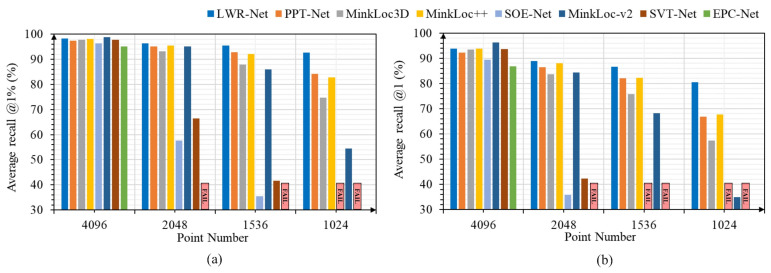
Robustness to the variances in the point density on the Oxford robotcar dataset: (**a**) average recall @1% changes with point density; (**b**) average recall @1 changes with point density.

**Figure 4 sensors-23-08664-f004:**
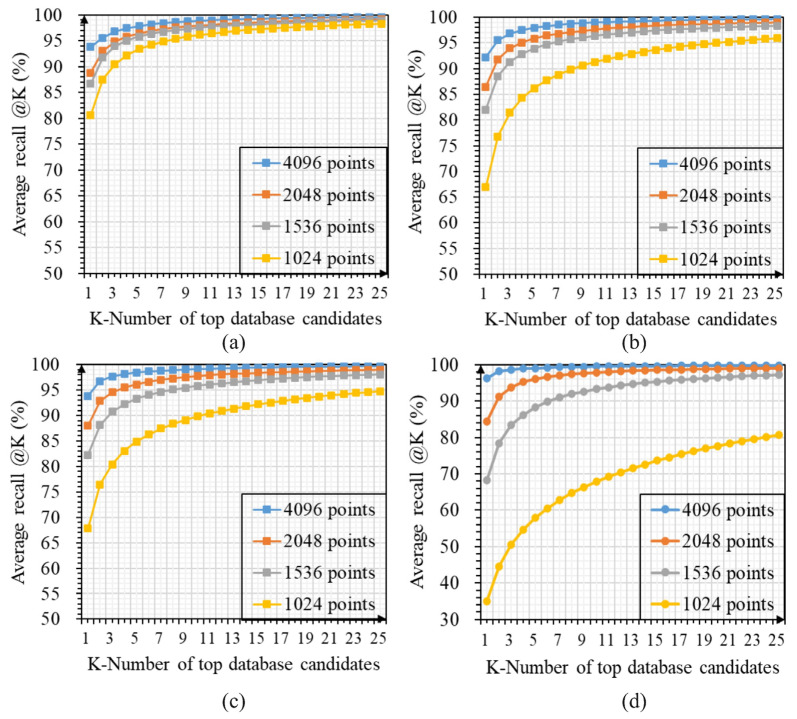
Changes in average recall curves of each model for the top 25 matches under the influence of different point density settings: (**a**) LWR-Net; (**b**) PPT-Net; (**c**) MinkLoc++; (**d**) MinkLoc-v2.

**Figure 5 sensors-23-08664-f005:**
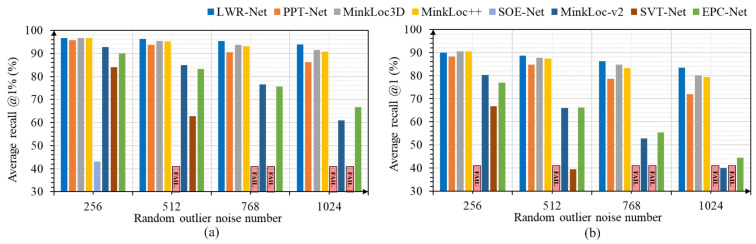
Robustness to the random outlier noises on the Oxford robotcar dataset: (**a**) average recall @1% changes with random outlier noise intensity; (**b**) average recall @1 changes with random outlier noise intensity.

**Figure 6 sensors-23-08664-f006:**
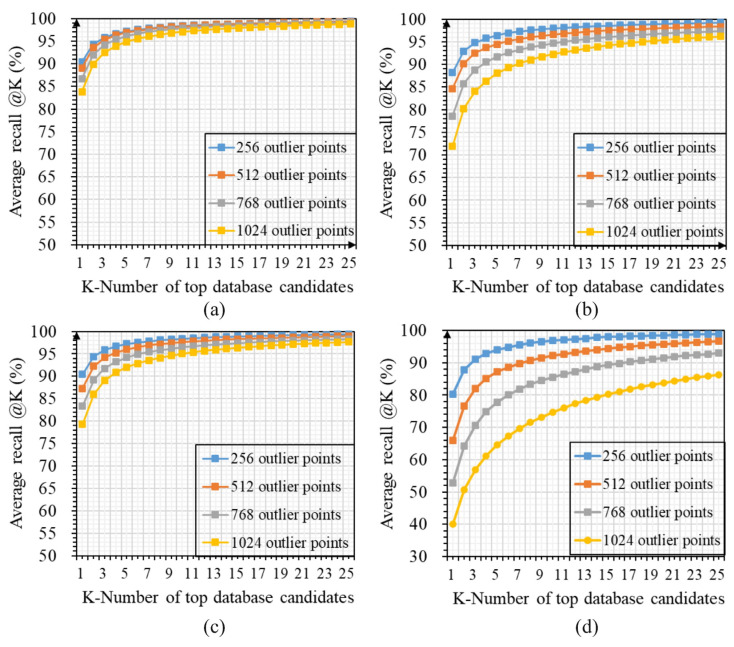
Changes in average recall curves of each model for the top 25 matches under the influence of different random outlier noise intensities: (**a**) LWR-Net; (**b**) PPT-Net; (**c**) MinkLoc3D++; (**d**) MinkLoc-v2.

**Figure 7 sensors-23-08664-f007:**
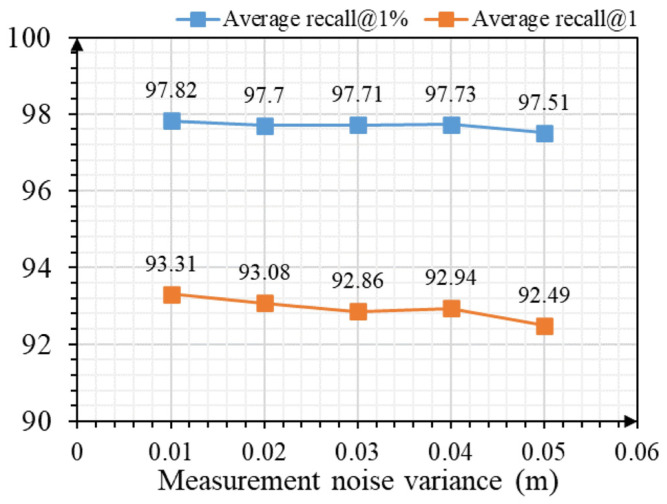
Recall indices change with measurement noise intensities.

**Figure 8 sensors-23-08664-f008:**
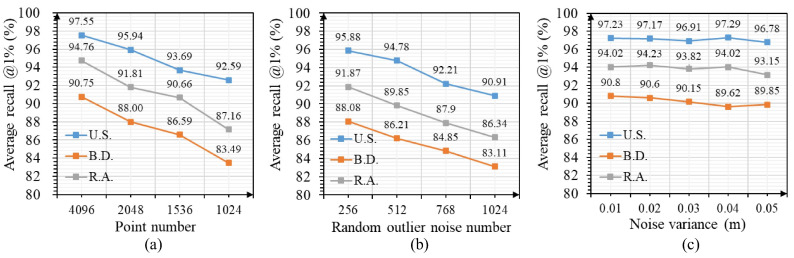
Robustness test results on three in-house datasets. (**a**) average recall @1% changes with point density; (**b**) average recall @1% changes with random outlier noise intensities; (**c**) average recall @1% changes with measurement noise intensities.

**Figure 9 sensors-23-08664-f009:**
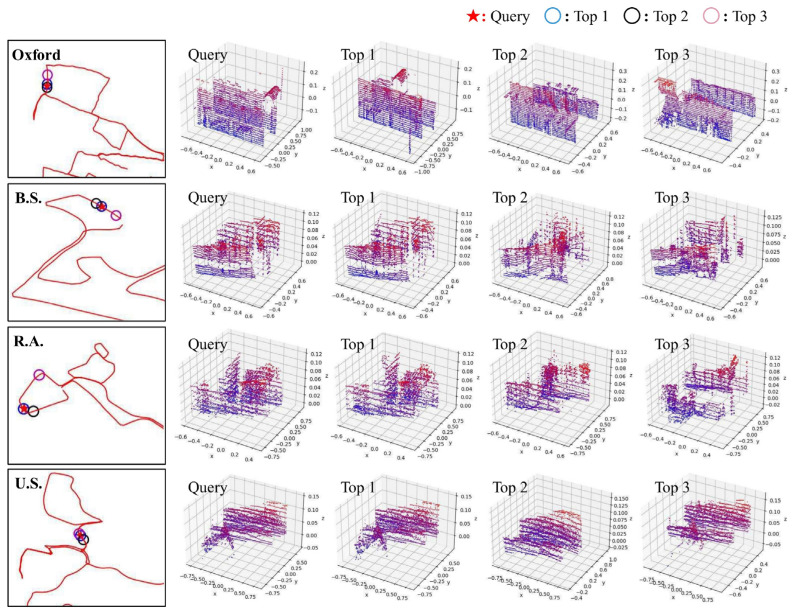
Place recognition results of LWR-Net on Oxford robotcar and three in-house datasets.

**Figure 10 sensors-23-08664-f010:**
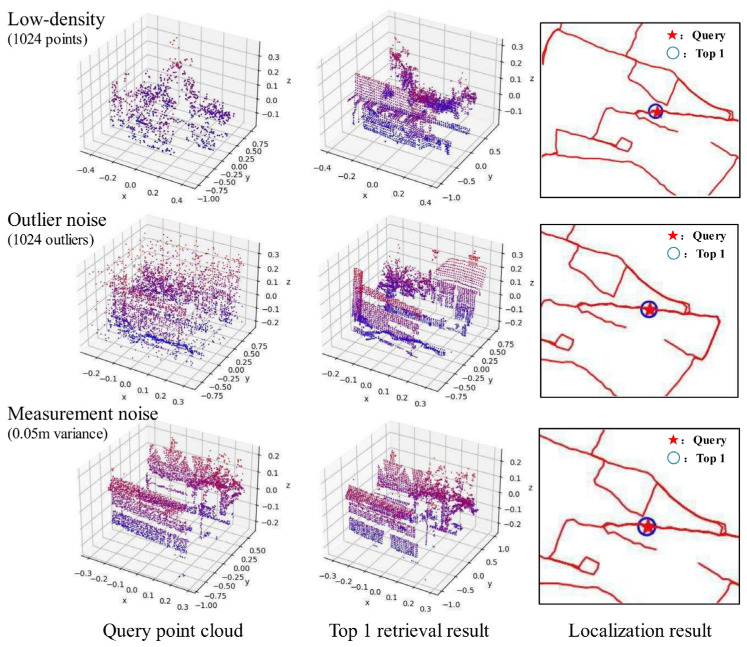
Results of LWR-Net in low-density and noisy conditions.

**Table 1 sensors-23-08664-t001:** Test on the standard Oxford robotcar dataset.

Model	Average Running Time	Model Size (Million)	Recall @1 (%)	Recall @1% (%)
PointNetVLAD	23 ms	19.78	62.76	81.01
PCAN	45 ms	20.42	69.05	83.81
LPD-Net	29 ms	19.81	84.19	94.44
SOE-Net	21 ms	19.40	89.37	96.40
EPC-Net	26 ms	4.70	86.84	95.19
MinkLoc3D	12 ms	1.06	93.48	97.85
NDT-Transformer	-	20.67	93.80	97.65
Minkloc++	12 ms	1.06	93.90	98.15
PPT-Net	21 ms	13.39	92.21	97.50
SVT-Net	13 ms	0.94	93.70	97.80
ASVT-Net	11 ms	**0.44**	93.90	98.00
MinkLoc-v2	19 ms	2.66	**96.25**	**98.87**
LWR-Net	**10 ms**	**0.44**	93.76	98.32

**Table 2 sensors-23-08664-t002:** Test on three in-house datasets using the trained model on the Oxford robotcar dataset.

Model	U.S		R.A.		B.D.	
	Recall @1 (%)	Recall @1% (%)	Recall @1 (%)	Recall @1% (%)	Recall @1 (%)	Recall @1% (%)
PointNetVLAD	63.01	77.83	56.19	69.76	57.21	65.30
PCAN	62.50	79.05	57.00	71.18	58.15	66.82
LPD-Net	74.95	91.93	73.12	84.28	74.33	83.20
SOE-Net	82.47	93.17	82.93	91.47	83.34	88.45
EPC-Net	86.40	95.43	79.45	88.11	77.77	84.40
MinkLoc3D	86.21	94.91	82.99	91.94	79.84	86.70
Minkloc++	86.08	94.52	83.73	92.06	82.38	88.43
PPT-Net	89.68	**97.93**	86.76	93.28	83.99	89.23
SVT-Net	90.10	96.50	84.30	92.70	85.50	90.70
ASVT-Net	87.90	96.10	83.30	92.00	82.30	88.40
MinkLoc-v2	90.85	96.65	86.49	93.75	85.26	90.15
LWR-Net	**91.16**	97.81	**88.31**	**94.89**	**85.71**	**90.75**

**Table 3 sensors-23-08664-t003:** Ablation study on the Oxford robotcar dataset.

Model	Model Size (Million)	Recall @1 (%)	Recall @1% (%)
LWR-Net	0.44	93.76	98.32
LWR-Core	0.30	93.08	97.65
LWR-WithoutRes	0.32	79.11	91.77
LWR-WithoutResLocal	0.16	78.13	91.84
LWR-WithoutResGlobal	0.16	75.52	89.57
LWR-WithoutDS	0.44	92.88	97.59

## Data Availability

The data sources used in this article are from the publicly available datasets. Readers can find relevant information about the data and download links at https://drive.google.com/drive/folders/1Wn1Lvvk0oAkwOUwR0R6apbrekdXAUg7D (accessed on 18 October 2023).

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
