# Peer review of "LWR-Net: Robust and Lightweight Place Recognition Network for Noisy and Low-Density Point Clouds"

_sensors, 2023, doi:10.3390/s23218664_

Round 1

Reviewer 1 Report

This paper proposed a lightweight place recognition network for efficient and robust point cloud retrieval in noisy, low-density conditions. The motivation and the proposed method are clearly described. The experimental results show that the proposed method can consistently outperform state-of-the-art methods.

There are some comments to improve its quality:

Is there a problem with the last name of the 4th author?

Is there a comma after the 2nd affiliate?

What is the dilated ratio $\sigma$ set to in the experiment?

The sequence in Figure 2 is: ...->basic encoding->local residual blocks->grouped self-attention->global residual blocks->... Equation (1) describes the whole process above? The Residual-MLP encoding module in the paper corresponds to the above process? Therefore, Attentional weighting module belongs to Residual-MLP encoding module. Please reorganize or rewrite this part.

The symbol G is used for the final descriptor in equation (5) and f is used in equation (6), so why not be consistent?

All figures should be refined in high resolution.

Some recent models are missing: Coarse-to-fine pipeline for 3D wireframe reconstruction from point cloud, Pointwise geometric and semantic learning network on 3D point clouds.

Reviewer 2 Report

Overall, this was a very well-written paper. The authors present a sound methodology, and present transparent results compared to the current state of the art of localization using 3D data. Furthermore, the contributions are clear: smaller model size and comparable accuracy on sparse data. One point to note is that the authors downsampled more wholesome data to generate sparse datasets, and accordingly in some ways the sparse data were partially ideal, as compared to truly sparse data as captured by a sensor. Additionally, some of the methods read as heuristic, though given the strong comparison to the existing state of the art, I do not see this as an issue.

The use of a standard public dataset, as well as three hand-crafted but comprehensive datasets, provides good methodological robustness. Will the in-house datasets be made public for future evaluation? 

Overall, I think this paper presents a reasonable contribution, though the importance of 3D localization techniques is not overly well justified, and might impact the interest of Sensor's readers.  

Reviewer 3 Report

The paper is well-written and relatively easy to follow. The proposed method is described with sufficient details and is technically sound. 

One suggestion for improvement is to include the recent works on point cloud-based place recognition:

- Z. Fan et al., "SVT-Net: Super Light-Weight Sparse Voxel Transformer for Large Scale Place Recognition" 2022 AAAI Conference presents a very lighweight model ASVT-Net (0.4 milion parameters) with 93.9 with Recall@1 and 98.0 Recall@1% on Oxford
- J. Komorowski "Improving Point Cloud Based Place Recognition with Ranking-based Loss and Large Batch Training" 2022 ICPR has 96.3 Recall@1 and 98.9 Recall@1% on Oxford
Another recent related work:
- T. Ye, Efficient 3D Point Cloud Feature Learning for Large-Scale Place Recognition, IEEE TRANSACTIONS ON INSTRUMENTATION AND MEASUREMENT, VOL. 71, 2022

The paper is relatively well-written and easy to follow. I've spotted one mistake:

Line 120 says "adversely affecting 120 the perceptron of the scene.". It should be "perception of the scene"
